# ME2 Promotes Hepatocellular Carcinoma Cell Migration through Pyruvate

**DOI:** 10.3390/metabo13040540

**Published:** 2023-04-10

**Authors:** Yanting Yang, Zhenxi Zhang, Wei Li, Li Li, Ying Zhou, Wenjing Du

**Affiliations:** 1State Key Laboratory of Medical Molecular Biology, Haihe Laboratory of Cell Ecosystem, Department of Cell Biology, Institute of Basic Medical Sciences Chinese Academy of Medical Sciences, School of Basic Medicine Peking Union Medical College, Beijing 100005, China; 2Key Laboratory of Cellular Physiology at Shanxi Medical University, Ministry of Education, and the Department of Physiology, Shanxi Medical University, Taiyuan 030606, China

**Keywords:** malic enzyme 2, pyruvate, cell migration and invasion

## Abstract

Cancer metastasis is still a major challenge in clinical cancer treatment. The migration and invasion of cancer cells into surrounding tissues and blood vessels is the primary step in cancer metastasis. However, the underlying mechanism of regulating cell migration and invasion are not fully understood. Here, we show the role of malic enzyme 2 (ME2) in promoting human liver cancer cell lines SK-Hep1 and Huh7 cells migration and invasion. Depletion of ME2 reduces cell migration and invasion, whereas overexpression of ME2 increases cell migration and invasion. Mechanistically, ME2 promotes the production of pyruvate, which directly binds to β-catenin and increases β-catenin protein levels. Notably, pyruvate treatment restores cell migration and invasion of ME2-depleted cells. Our findings provide a mechanistic understanding of the link between ME2 and cell migration and invasion.

## 1. Introduction

Cancer metastasis is the leading cause of cancer-related death [1]. The migration and invasion of cancer cells are the initial steps in metastasis [2]. Cell migration plays an important role in embryonic development, cell homeostasis, wound healing, and other pathological states [3]. Blocking the ability of cancer cells to migrate and invade offers a new approach to treat patients with malignant diseases [4]. It has been reported that activation of multiple oncogenic signals, such as ERK, JNK, Wnt/β-catenin pathways, etc., can induce hepatocellular carcinoma migration [5,6,7].

Cellular metabolism remodeling is now recognized as one of the hallmarks of cancer [8]. Cancer metabolism plays a potential role in driving cancer cell phenotype and promoting cancer aggressiveness during tumor progression [9,10]. Increasing evidence suggests that the abnormal expression of metabolic enzymes significantly affects the migration and invasion of tumor cells, such as FBP1 and LDH [11,12]. It was reported by Dong et al. that loss of FBP1 promotes the progression of EMT and basal-like breast cancer [11]. For LDH5, as in Koukourakis’ study, they found that high LDH5 content in tumor cells was associated with metastasis and aggressiveness of colorectal adenocarcinoma [12]. Targeting cancer metabolism is an attractive strategy to limit metastasis [13]. However, the underlying mechanisms of how cancer metabolism regulates tumor cell migration still need further investigation.

Malic enzymes are involved in regulating glutamine metabolism by catalyzing the oxidative decarboxylation of malate to pyruvate and NADPH [14]. Malic enzymes have multiple functions in cells, including cell growth, senescence, and adipogenesis [14,15]. There are three isoforms of malic enzymes in mammals, ME1, ME2, and ME3. ME2 is the major isomer with the highest abundance and enzyme activity in mitochondria [16]. Overexpression of ME2 has also been associated with cell migration and invasion [17,18]. However, the potential mechanisms by which ME2 regulates cancer cell migration remain unclear.

Here, we identified the role of ME2 in regulating liver cancer cells migration via increasing the pyruvate production, which binds to and increases β-catenin protein stability, thereby promoting cell migration and invasion. These findings provide a mechanistic understanding of the link between ME2 and cell migration.

## 2. Materials and Methods

### 2.1. Antibodies and Reagents

Anti-ME2 (ab139686; dilution: 1/1000) was purchased from Abcam (Cambridge, UK). Antibodies of β-actin (66009-1; dilution: 1/3000) and β-tubulin (10094-1-AP; dilution: 1/3000) were from Proteintech (Chicago, IL, USA). The anti-β-catenin (8480; dilution: 1/1000) was ordered from Cell Signaling Technology (Beverly, MA, USA). Dimethyl pyruvate (371173) was purchased from Sigma-Aldrich (St. Louis, MO, USA).

### 2.2. Cell Culture and Transfection

Human liver cancer cell lines SK-Hep1 and Huh7 were obtained from the Cell Resource Center, the Shanghai Institute of Life Sciences (Shanghai, China). SK-Hep1 and Huh7 cells were cultured in standard Dulbecco’s modified Eagle’s medium (DMEM, Life Technologies). The DMEM mediums supplemented with 10% fetal bovine serum (FBS). All cells were cultured in a 5% CO_2_ humidified incubator (Thermo Fisher Scientific, Waltham, MA, USA) at 37 °C. All the cell lines have been authenticated.

siRNAs were transfected using Lipofectamine RNAiMAX (Life Technologies, Invitrogen, Carlsbad, CA, USA) following the manufacturer’s instruction. The following siRNAs were ordered from GenePharma Company (Suzhou, China):

ME2#1: 5′-GUCGACAUUUGCACAUAAATT-3′;

ME2#2: 5′-CACGGCUGAAGAAGCAUAUTT-3′;

PKM2: 5′-CGAUCAGUGGAGACGUUGAAG-3′;

ShRNAs expression plasmid were prepared in a pLKO.1-puro vector. The target sequences were as follows: ME2-1: 5′-GCAAGCCACTTATGCTGAACC-3′; ME2-2: 5′-GCACGGCTGAAGAAGCATATA-3′. Stable shRNA transfections were selected in medium containing puromycin (Calbiochem, San Diego, CA, USA, catalog No: 540222).

### 2.3. Western Blotting

Cells were seeded into 6-well plates at a density of 2 × 10^5^ cells per well and were incubated for 24 h for attachment. Then, the cells were transfected with ME2 siRNA (80 pmol) with or without 1 mM dimethyl-pyruvate for 48 h. Whole cell lysates were prepared in RIPA lysis buffer containing a cocktail of protease inhibitors for 30 min on ice, protein quantitation was performed using BCA assay, and they were then boiled. Total cellular proteins were electrophoresed by SDS-PAGE and then probed with the indicated antibodies.

### 2.4. Quantitative PCR Analysis

Total RNA was isolated from cells by RNAsimple Total RNA Kit (TIANGEN, Beijing, China). After being reversed to cDNA using a First-strand cDNA Synthesis Kit (TIANGEN), a CFX96 Real-Time PCR system (Bio-Rad, Santa Clara, CA, USA) was used for quantitative PCR, and the amplifications were performed using SYBR Green PCR Mix. Statistical analysis was performed using GraphPad Prism (v8.0). The level of gene expression was normalized to internal control gene. The primer pairs were as follows:

h*GAPDH*: F, 5′- CCATGGGGAAGGTGAAGGTC-3′;

R, 5′- GCGCCCAATACGACCAAATC-3′;

h*ME2*: F, 5′- ATATACACCGACGGTTGGTCT-3′;

R, 5′- CATCAGTCACTACAACAGCCTT-3′;

h*CTNNB1*: F, 5′-CTGAGGAGCAGCTTCAGTCC-3′;

R, 5′- GGCCATGTCCAACTCCATCA-3′;

h*PKM2*: F, 5′- TCTCTTCGTCTTTGCAGCGT-3′;

R, 5′- CAGCTGCTGGGTCTGAATGA-3′;

### 2.5. LC-MS Analysis of Cell Metabolites

For intracellular metabolite extraction, cells were washed 3 times with PBS. Then, 1 mL of ice-cold 80% HPLC-grade methanol was added to a 1.5 mL centrifuge tube containing cells and placed on dry ice for 30 min. The tubes were incubated on ice for 20 min, then centrifuged at 15,000× *g* for 15 min at 4 °C. The supernatant containing cellular metabolites was transferred into a clean 1.5 mL centrifuge tube. Metabolite extraction was repeated again. The aqueous phase was dried with a nitrogen flow evaporator. The protein pellet was resuspended in 100 μL of 1% SDS buffer for BCA assays. The extracts were analyzed with a Triple Quadrupole LC/MS System (Agilent, 1290/6460, Santa Clara, CA, USA).

### 2.6. Measurement of Pyruvate

Cells were seeded in a 6-well plate of 2 × 10^5^ cells per well. After 24 h, the cells were transfected with ME2 siRNA (80 pmol) with or without 1 mM dimethyl-pyruvate for 48 h. Pyruvate levels were analyzed by pyruvate assay kit (ab65342, Abcam, Cambridge, UK), following the manufacturer’s instructions.

### 2.7. Wound Healing Assay

Cells were cultured overnight in a 12-well plate of 1 × 10^5^ cells per well. When the cell fusion was around 80%, a scrape wound was created with the tip of a sterile 10 μL pipette [19]. The medium was replaced by serum-free DMEM, and cells continued to be cultured for 24 h [5,20,21]. After 24 h, the pictures were taken at the same position and the relative distance the cells had moved was calculated. The experiment was repeated 3 times and the average value was calculated.

### 2.8. Transwell Assay

Then, 1.5 × 10^3^ cells were seeded into the upper surface of a 24-well Transwell unit (Corning). DMEM containing 10% FBS was added in the medium. Meanwhile, 750 µL medium supplied with 20% FBS was added to the lower chamber. The plates were incubated at 37 °C. After 24 h, the units were washed with PBS three times before removing the upper cells. Subsequently, migrated cells were immobilized with methanol and stained with 0.1% crystal violet. Migrated cells were pictured.

### 2.9. Drug Affinity Responsive Targets Stability (DARTS)

DARTS was performed as previously described [22]. Briefly, 2 × 10^5^ cells were lysed in M-PER containing a cocktail of protease inhibitors and phosphatase inhibitors. TNC buffer (10 mM CaCl_2_, 50 mM Tris-HCL pH8.0, and 50 mM NaCl) was added to the lysate. Cell lysates were added with various concentrations of pyruvate or ddH_2_O (vehicle) for 1 h at room temperature and were then digested with pronase (1:3000 dilution for β-catenin) for 30 min. The digestion was stopped by boiling. Western blotting was used to test whether β-catenin is a direct target of pyruvate. β-tubulin was used as a negative control.

### 2.10. Cellular Thermal Shift Assay (CETSA)

CETSA measures the direct binding between pyruvate and β-catenin in cells, as previously de-scribed [23]. Briefly, cells were cultured in a 6-cm dish at a density of 5 × 10^5^ cells. After 24 h, cells were pretreated with 1 mM dimethyl-pyruvate for 8 h. Cells were resuspended with PBS containing a protease inhibitor cocktail. The cell suspension was heated to denaturing proteins in a Bio-Rad T100 Thermal Cycler at an indicated temperature for 3 min, and cooled at 25 °C for another 3 min. Finally, all the samples were subjected to three freeze-thaw cycles, and centrifuged at 20,000 g for 20 min at 4 °C. The supernatant was boiled for Western blotting. Quantified banding was conducted using ImageJ software (v2.0.2).

### 2.11. Molecular Docking

The docking studies were performed by AutoDock4 [24,25]. The structure of pyruvate was obtained from the chemical database. The structure of β-catenin was taken from the Protein Data Bank (PDB code: 2Z6H). The β-catenin was used as the receptor. AutoDock4 was used to prepare ligands and receptors for docking. The grid size was set to 40 × 40 × 40 points with a grid spacing of 1 Å around the center of the potential area. Each ligand was independently docked five times. The docking results were analyzed and visualized by PyMOL (v2.1).

### 2.12. Statistical Analysis

No statistical methods were used to predetermine sample size. Data are presented as mean± standard deviation. All statistical analyses were performed. A *p* value < 0.05 was considered to indicate statistical significance. Statistical significance is shown as * *p* < 0.05, ** *p* < 0.01, *** *p* < 0.001. Statistical analyses were performed using GraphPad Prism v.8.0.

## 3. Results

### 3.1. ME2 Promotes the Migration of HCC Cells

To evaluate the role of ME2 in cell migration, we performed wound-healing and transwell migration assays. We knocked down the expression of ME2 by using two different sets of small interfering RNA (siRNA) in human hepatocellular carcinoma SK-Hep 1 cells. Knockdown of ME2 significantly inhibited directional cell migration towards a “wound” in a cell monolayer (Figure 1A). Additionally, we stably knocked down ME2 in SK-Hep 1 cells using small hairpin RNA (shRNA). ME2-depletion significantly reduced the migration of SK-Hep 1 cells (Figure 1B). In contrast, overexpression of ME2 enhanced cell migration (Figure 1C). Consistent with the results in SK-Hep 1 cells, ME2-depletion also inhibited cell migration in human hepatocellular carcinoma Huh 7 cells (Figure 1D,E), whereas up-regulation of ME2 enhanced it (Figure 1F). Furthermore, in the transwell migration assay, depletion of ME2 markedly inhibited SK-Hep 1 cells migration through a permeable filter (Figure 2A,B), whereas overexpression of ME2 increased the migration (Figure 2C). Similar results were obtained in Huh 7 cells (Figure 2D–F). These data suggest that ME2 promotes cell migration and invasion.

### 3.2. ME2 Promotes β-Catenin Expression

We next investigated the mechanism by which ME2 promotes cell migration and invasion. We performed transcriptomic profiling of ME2-depleted cells by RNA sequencing. Gene Ontology (GO) enrichment analysis showed that the differentially expressed genes (DEGs) were mainly enriched in the Wnt signaling pathway, the cell-cell adhesion signaling pathway, and the cell division pathway (Figure 3A). Gene set enrichment analysis (GSEA) of the genome-wide dataset revealed that ME2 silence correlated with a decreased gene signature of the Wnt/β-catenin signaling pathway (Figure 3B). These results suggest that ME2 regulates the Wnt/β-catenin signaling pathway.

The Wnt/β-catenin pathway is one of the common signaling pathways controlling epithelial-mesenchymal transition (EMT) [26]. β-catenin is an indispensable structural component of cadherin-based cell-cell junctions, and the key nuclear effector of canonical Wnt signaling in the nucleus [27]. Therefore, we determined whether ME2 regulates β-catenin expression. We used two sets of siRNAs to knock down ME2 expression in SK-Hep 1 and Huh 7 cells. The expression of β-catenin mRNA and protein was detected by real-time PCR and Western blotting. Knockdown of ME2 had no effect on mRNA levels of β-catenin in SK-Hep 1 and Huh 7 cells (Figure 4A,B). However, the protein levels of β-catenin were decreased following ME2-silencing by siRNA or shRNA (Figure 4C–F). Conversely, enforced expression of ME2 augmented β-catenin expression (Figure 4G,H). Moreover, introducing an siRNA-resistant form of ME2 in ME2-siRNA-treated SK-Hep1 and Huh7 cells largely restored the β-catenin protein levels (Figure 4I,J). These results suggest that ME2 promotes β-catenin expression.

### 3.3. ME2 Promotes β-Catenin Expression via Pyruvate

We next investigated the mechanism by which ME2 regulates the expression of β-catenin. ME2 plays a key role in the TCA cycle by catalyzing the conversion of malate to pyruvate (Figure 5A). Thus, we first examined the levels of malate and pyruvate in the control or ME2 knockdown SK-Hep1 cells by LC-MS. As expected, ME2 silencing resulted in a decrease of its product pyruvate and an increase in its substrate malate (Figure 5B). Moreover, the addition of pyruvate restored intracellular pyruvate concentration in ME2-depleted SK-Hep1 and Huh7 cells (Figure 5C,D). Consistent with this, the addition of pyruvate also restored the expression of β-catenin in ME2-depleted cells (Figure 5E–H). Taken together, these results indicate that ME2 regulates β-catenin expression by modulating cellular pyruvate levels.

Furthermore, to manipulate cellular pyruvate levels, we also knocked down the expression of PKM2, one of the key enzymes of glycolysis, which catalyzes the conversion of phosphoenolpyruvate to pyruvate. Consistent with the results of knocking down ME2, the depletion of PKM2 resulted in reduced expression of β-catenin in SK-Hep1 and Huh7 cells (Figure 6A,B), and, the addition of pyruvate restored β-catenin expression (Figure 6C,D). Taken together, these results indicate that cellular pyruvate levels could regulate β-catenin expression.

### 3.4. Pyruvate Directly Binds to β-Catenin and Inhibits Protein Degradation

We further investigated the mechanism by which pyruvate promotes β-catenin expression. We first examined whether pyruvate binds to the β-catenin. Drug affinity responsive target stability (DARTS) analysis suggests that pyruvate binds to and protects β-catenin from pronase degradation in a dose-dependent manner (Figure 7A,B). Moreover, in cellular thermal shift assays (CETSAs), the melting curve of β-catenin was shifted in the presence of pyruvate (Figure 7C,D). To further evaluate a certain binding site, a molecular docking simulation was performed using a reported β-catenin structure (PDB code: 2Z6H) by AutoDock4. The predicted mode revealed that pyruvate Arg515, His578, Arg612 might be able to anchor pyruvate in the pocket by forming hydrogen bonds with pyruvate (Figure 7E). Then, we examined the stability of β-catenin using cycloheximide (CHX) chase assay. Down-regulation of ME2 expression accelerates degradation of β-catenin in SK-Hep 1 cells, and notably, supplying cells with pyruvate restored the stability of β-catenin in ME2-depleted cells (Figure 7F). These results indicate that pyruvate binds to and stabilizes β-catenin.

### 3.5. Pyruvate Supplementation Restores Cell Migration Ability in ME2-Depleted HCC Cells

We next explored the role of pyruvate in ME2-mediated cell migration and invasion. Consistent with previous data, ME2 depletion inhibited directional cell migration toward a “wound” in a cell monolayer in SK-Hep 1 cells, while supplementation with pyruvate significantly restored it (Figure 8A,B). Similar results were obtained in Huh 7 cells (Figure 8C,D). Moreover, silencing of ME2 by siRNA or shRNA markedly inhibited cell migration through a permeable filter in SK-Hep1 and Huh7 cells, while treatment of pyruvate restored it (Figure 8E–H). Taken together, these results suggest that ME2 promotes cell migration and invasion at least partially through pyruvate.

### 3.6. The Interaction between ME2 and β-Catenin

We next explored the interaction between ME2 and β-catenin. Firstly, we conducted an interactome analysis to place our results in what is globally known regarding the molecular relationships of catenin in human and cancer metabolisms. We predicted proteins known to be involved in the β-catenin (CTNNB1) complex and metabolism using STRING. In terms of results, as shown in Figure 9A, the network calculated by STRING showed that β-catenin, as its central HUB node, does not directly interact with ME2. Furthermore, the correlation between ME2 and β-catenin was analyzed in a liver hepatocellular carcinoma cohort based on the GEPIA database. To some extent, elevated ME2 expression was consistent with the increased expression of catenin (R = 0.33) (Figure 9B). These data indicated that ME2 could regulate catenin to some certain extent.

## 4. Discussion

ME2 is involved in the TCA cycle and regulates the glutamine metabolism in some cancer cells [14,28]. ME2 plays an important role in regulating the cell growth, proliferation, and invasion of a variety of tumor cells [14,17,18,29]. Although ME2 expression has been found to influence the migration and invasion of melanoma cells, this function is related to the energy supply for cell mobility [17]. In addition, ME2 also regulates insulin secretion in INS-1 cells through pyruvate cycling [30]. ME2 plays a crucial role in modulating lung cancer differentiation and growth [31]. In this study, we show the role of ME2 in promoting the migration and invasion of hepatocellular carcinoma cells. Notably, this function of ME2 is mediated by its product pyruvate, which directly binds to β-catenin and increases β-catenin protein levels. Our recent study reported that pyruvate directly binds to and activates acetaldehyde dehydrogenase 2 (ALDH2) [32]. Together, these findings indicate that pyruvate may have a broader range of new functions within cells in a metabolism-independent manner.

As a metabolic intermediate in the glycolytic pathway, part of the TCA cycle, pyruvate is commonly added to the cell culture medium as a growth supplement. Sodium pyruvate is also widely used as a food supplement [33]. Pyruvate has a dual role as an energy provider (such as ATP formation) and an antioxidant ability (cellular respiration) [34,35]. The antioxidant effect gives pyruvate cytoprotective capabilities. However, our study reveals a previously unrecognized and metabolically independent role for pyruvate in the control of cell migration and invasion by acting as an endogenous regulator of β-catenin. Therefore, dietary pyruvate supplements are not beneficial for everyone. Conversely, the long-term uptake of pyruvate can worsen the health status of cancer patients.

As part of the E-cadherin complex, β-catenin is one of the key players in the regulation of cadherin-mediated cell-cell adhesion [36]. It is a transcriptional co-activator in the canonical Wnt signaling pathway [26]. The canonical Wnt/β-catenin pathway plays a critical role in the proliferation and metastasis of cancer cells [37]. In many cancers, Wnt/β-catenin signaling is constitutively active and promotes EMT [7,27]. Several systematic studies have found that β-catenin physically interacts with many other proteins to form complexes that activate myriad functions through proteomic profiling and genetic interaction mapping [38]. Therefore, there may be a molecular network formed by ME2 and β-catenin to achieve functional biological processes together. STRING is a database of known and predicted protein-protein interactions. There is indirect evidence that ME2, LDHA and LDHB have a functional link between these three proteins based on STRING [39,40]. There is evidence again in the Pathway Interaction Database of functional interactions between JUN and CTNNB1 or between EP300 and CTNNB1 [41]. We predicted proteins known to be involved in the β-catenin (CTNNB1) complex and metabolism using STRING. In terms of results, as shown in Figure 9A, β-catenin does not directly interact with ME2. This may be due to the fact that the software calculates the correlation is based on five data to predict published proteins interacting with β-catenin. However, ME2 and β-catenin have not been reported in the known literature, so the relationship between ME2 and β-catenin cannot be judged by software. However, the correlation between ME2 and β-catenin in the liver hepatocellular carcinoma cohort based on the GEPIA database showed that elevated ME2 expression was consistent with the increased expression of catenin (R = 0.33) (Figure 9B). These data indicated that ME2 could regulate catenin to a certain extent.

Although the mechanisms of the constitutive activation of Wnt/β-catenin signaling in cancer cells has been well studied, our study found that pyruvate, a metabolite of the glucose metabolism, increases β-catenin protein levels to promote cell migration and invasion. Cancer cells consume large amounts of glucose and rapidly convert it to pyruvate via glycolysis. Perhaps the accumulation of pyruvate is one reason for the sustained activation of Wnt/β-catenin signaling in cancer cells. Although further study is needed to fully understand exactly how pyruvate acts on β-catenin, our findings suggest that pyruvate may act as an endogenous β-catenin regulator. In other words, pyruvate may be a signaling molecule that activates the intracellular Wnt/β-catenin signaling pathway.

Collectively, our study reveals a previously unrecognized and metabolism-independent role of pyruvate in controlling cell migration and invasion. We think that pyruvate may be a signaling molecule that activates the Wnt/β-catenin signaling pathway. These findings provide a connection between metabolites and the regulation of cell migration. In terms of clinical significance, our study shows the risk of tumor growth inhibition with pyruvate supplementation in cancer patients. Supplementation with additional pyruvate may be harmful to patient recovery and significantly promote cancer metastasis.

## Figures and Tables

**Figure 1 metabolites-13-00540-f001:**
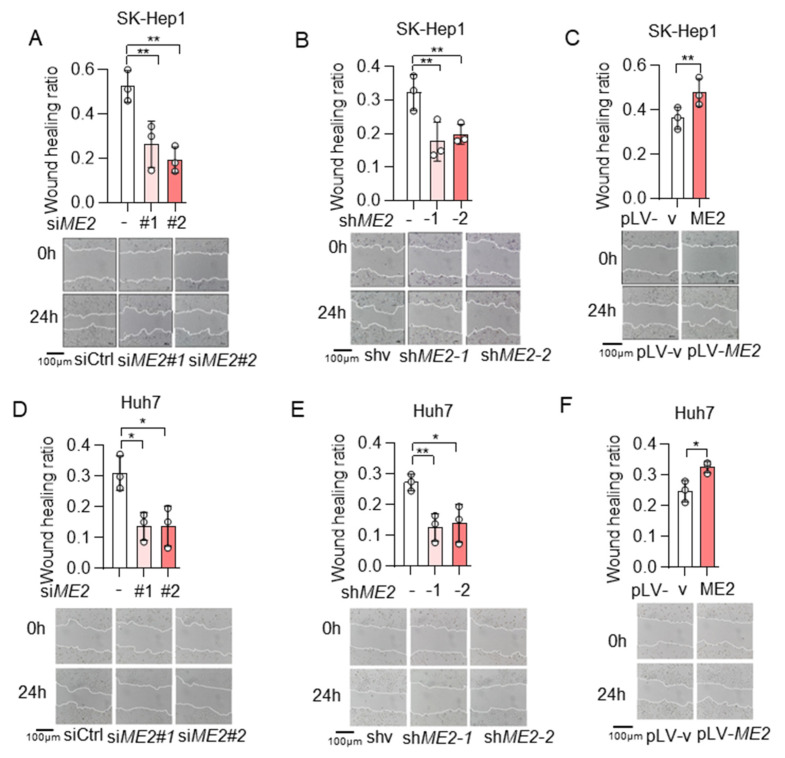
ME2 promotes cell wound healing migration in SK-Hep 1 and Huh 7 cells. (**A**,**B**) SK-Hep1 cells were transfected with ME2 siRNA (**A**) or stably infected with retroviral vectors expressing ME2 (**B**). (**C**) SK-Hep1 cells were stably over-expressed ME2 or vector control. (**D**,**E**) Huh7 cells were transfected with ME2 siRNA (**D**) or stably infected with retroviral vectors expressing ME2 (**E**). (**F**) Huh7 cells were stably over-expressed ME2 or vector control. Migratory properties were analyzed using wound healing assays. Statistical analysis of the cells show relative open wounds (top). Representative images are shown (bottom). Representative images are shown (bottom). Data in (**A**–**F**) are from n  =  3 biological independent wells. Data are the mean ± s.d. Statistical significance was determined by a two-tailed unpaired *t*-test. * *p* < 0.05, ** *p* < 0.01.

**Figure 2 metabolites-13-00540-f002:**
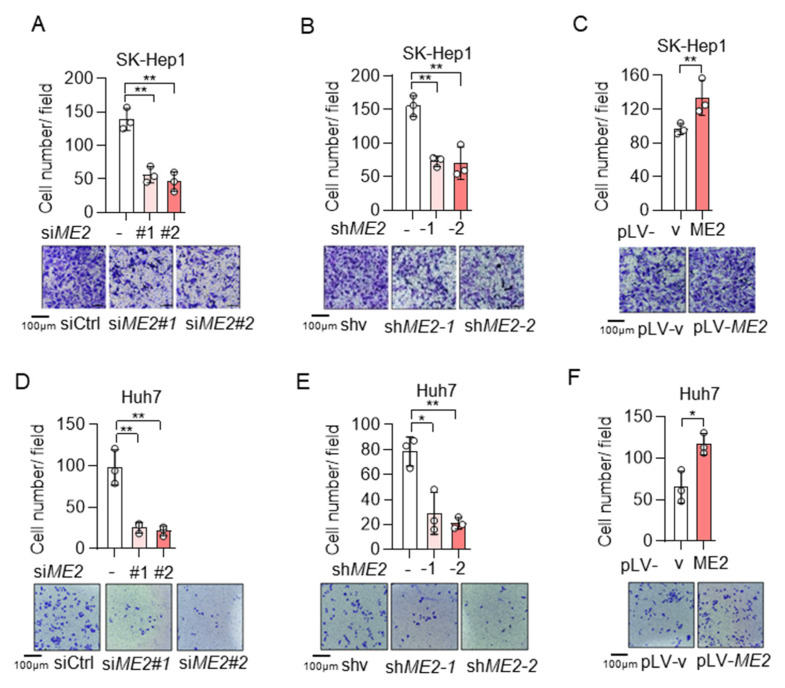
ME2 promotes cell migration and invasion abilities in SK-Hep 1 and Huh 7 cells. (**A**,**B**) SK-Hep1 cells were transfected with ME2 siRNA (**A**) or stably infected with retroviral vectors expressing ME2 (**B**). (**C**) SK-Hep1 cells were stably over-expressed ME2 or vector control. (**D**,**E**) Huh7 cells were transfected with ME2 siRNA (**D**) or stably infected with retroviral vectors expressing ME2 (**E**). (**F**) Huh7 cells were stably over-expressed ME2 or vector control. Migration was assessed by transwell. Numbers of migrated cells in the bar graph were quantified (top). Representative images are shown (bottom). Data in (**A**–**F**) are from n  =  3 biological independent wells. Data are the mean ± s.d. Statistical significance was determined by a two-tailed unpaired *t*-test. * *p* < 0.05, ** *p* < 0.01.

**Figure 3 metabolites-13-00540-f003:**
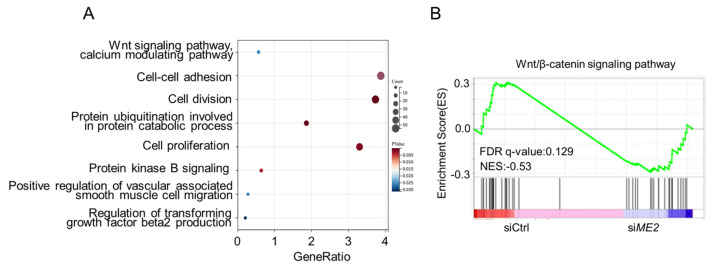
ME2 regulates the Wnt/β-catenin signaling pathway. (**A**) Gene Ontology (GO) enrichment analysis of the target genes from RNA-seq data in ME2 silenced cells. (**B**) Gene Set Enrichment Analysis (GSEA) from RNA-Seq datasets prioritizes the pathway associated with Wnt/β-catenin in ME2 silenced cells.

**Figure 4 metabolites-13-00540-f004:**
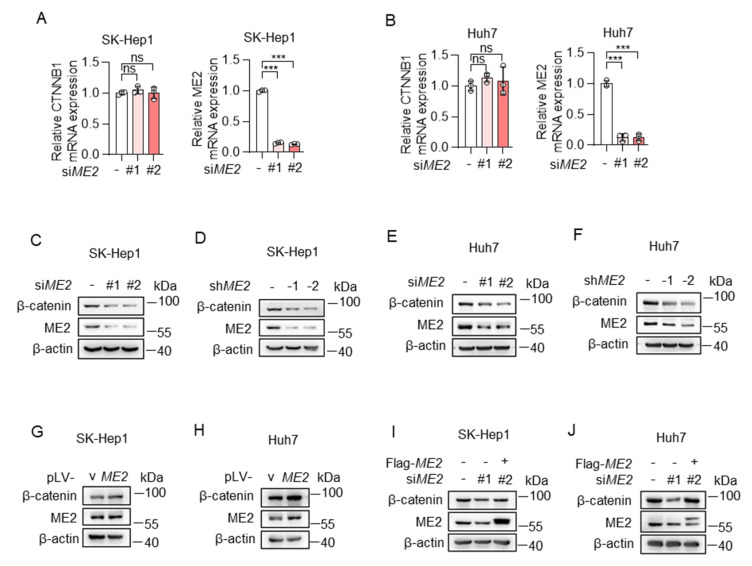
ME2 promotes β-catenin expression. (**A**,**B**) SK-Hep1 cells (**A**) and Huh7 cells (**B**) transfected with control or ME2 siRNAs. mRNA levels of CTNNB1 (left) and ME2 (right) were determined by qRT-PCR. (**C**–**F**) SK-Hep1 (**C**,**D**) and Huh7 (**E**,**F**) cells were transfected with ME2 siRNAs (**C**,**E**) or stably infected with retroviral vectors expressing ME2 (**D**,**F**). β-catenin protein expression is shown. (**G**,**H**) SK-Hep1 (**G**) and Huh7 (**H**) cells were stably over-expressed ME2 or vector control. β-catenin protein expression is shown. (**I**,**J**) SK-Hep1 (**I**) and Huh7 (**J**) cells were transfected with ME2 siRNA in the presence of ME2 cDNA. β-catenin protein expression is shown. Data in (**A**,**B**) are from n = 3 technical replicates from one of three independent experiments with similar results. Data are the mean ± s.d. Statistical significance was determined by two-tailed unpaired *t*-test. Western blots are representative of three independent experiments. ns *p* > 0.05, *** *p* < 0.001.

**Figure 5 metabolites-13-00540-f005:**
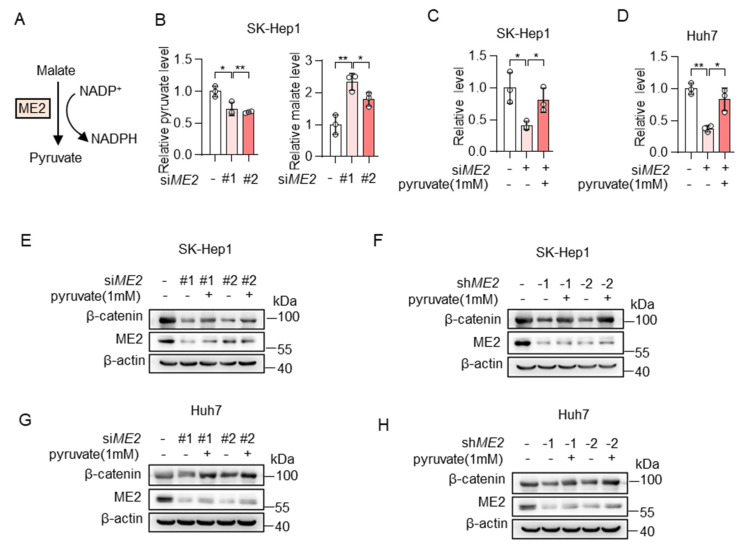
ME2 promotes β-catenin expression via pyruvate. (**A**) Model illustrating the catalytic mechanism of ME2. (**B**) Metabolites of control or ME2-knocked down SK-Hep1 cells were measured by LC-MS. (**C**,**D**) SK-Hep1 (**C**) and Huh7 (**D**) cells expressing ME2 siRNA were treated with or without dimethyl pyruvate (1 mM) for 48 h. Relative pyruvate levels were determined. (**E**–**H)** SK-Hep1 (**E**,**F**) and Huh7 (**G**,**H**) cells expressing ME2 siRNA (**E**,**G**) or ME2 shRNA (**F**,**H**) were treated with dimethyl pyruvate (1 mM) for 48 h. β-catenin protein expression is shown. Data in (**B**–**D**) are from n = 3 biological replicates. Data are the mean ± standard deviation (s.d.). Statistical significance was determined by two-tailed unpaired *t*-test. Western blots are representative of three independent experiments. * *p* < 0.05, ** *p* < 0.01.

**Figure 6 metabolites-13-00540-f006:**
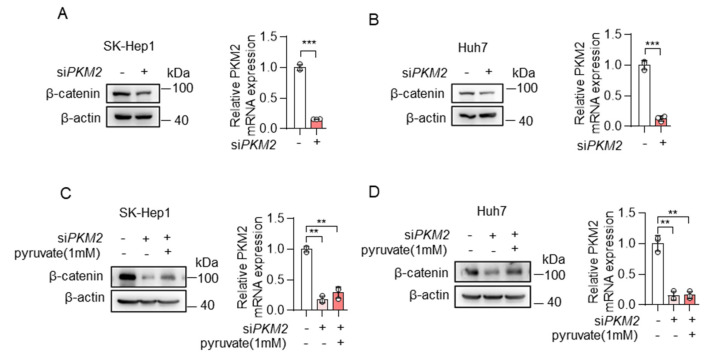
Pyruvate promoted the expression of β-catenin. (**A**,**B**) SK-Hep1 cells (**A**) and Huh7 cells (**B**) were transfected with PKM2 siRNA. β-catenin protein (left) and PKM2 mRNA (right) expression are shown. (**C**,**D**) SK-Hep1 cells (**C**) and Huh7 cells (**D**) were transfected with PKM2 siRNA in the presence or absence of dimethyl pyruvate (1 mM) for 48 h. β-catenin protein (left) and PKM2 mRNA (right) expression are shown. Data in (**A**–**D**) (right) are from n = 3 technical replicates from one of three independent experiments with similar results. Data are the mean ± standard deviation (s.d.). Statistical significance was determined by two-tailed unpaired *t*-test. Western blots are representative of three independent experiments. ** *p* < 0.01; *** *p* < 0.001.

**Figure 7 metabolites-13-00540-f007:**
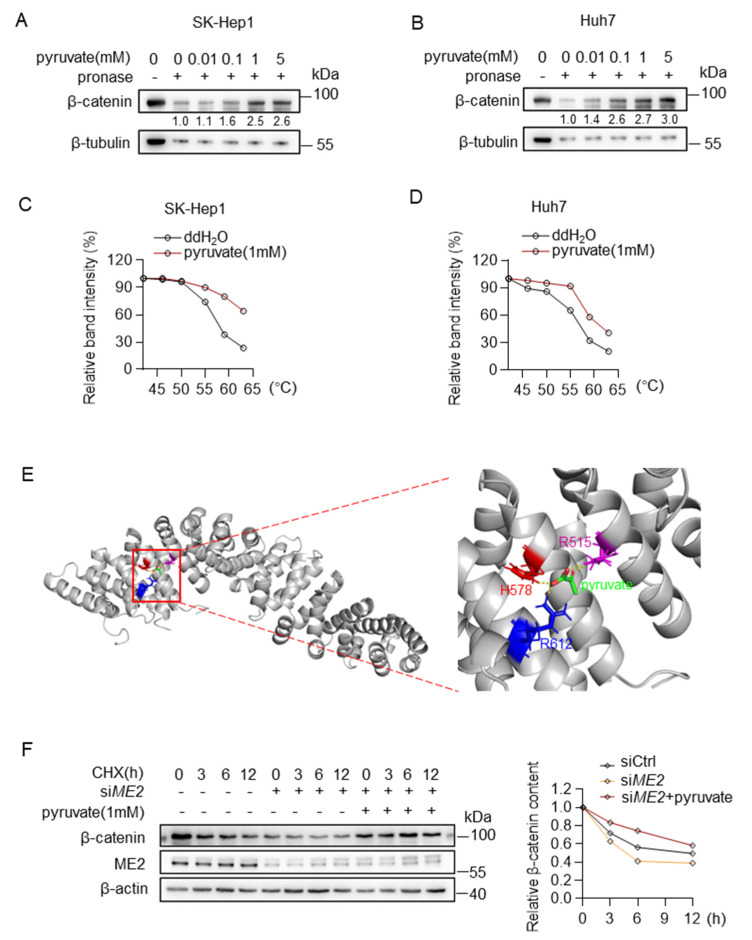
Pyruvate directly binds to β-catenin and inhibits protein degradation. (**A**,**B**) Direct binding of pyruvate and β-catenin was identified by DARTS assay in SK-Hep1 cells (**A**) and Huh7 cells (**B**). (**C**,**D**) CETSAs exhibit the binding affinity of pyruvate to β-catenin in SK-Hep1 cells (**C**) and Huh7 cells (**D**). (**E**) The predicted structure of β-catenin binding with pyruvate through AutoDock. Intermolecular interactions between pyruvate and its binding amino acid residues. (**F**) Western blotting analysis for β-catenin after treatment of 20 μg/mL of CHX in ME2-depleted SK-Hep1 cells with the presence or absence of dimethyl pyruvate (1 mM). Data in (**C**,**D**,**F**) (right) are from one of three independent experiments with similar results. In (**A**,**B**,**F**) (left), Western blots are representative of three independent experiments.

**Figure 8 metabolites-13-00540-f008:**
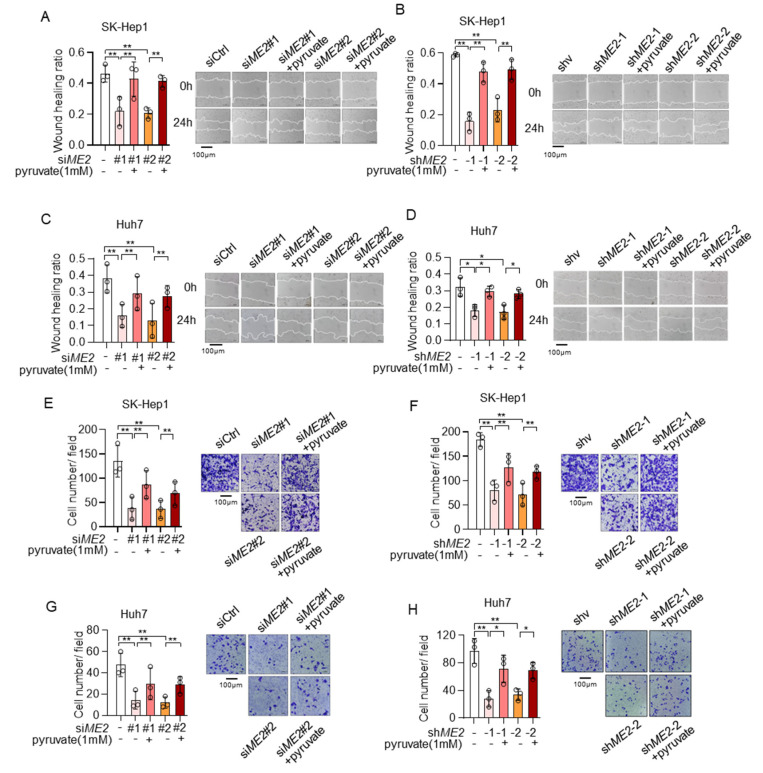
Pyruvate supplementation restores cell migration in ME2-depleted cells. (**A**–**D**)**,** SK-Hep1 (**A**,**B**) and Huh7 (**C**,**D**) cells expressing ME2 siRNA (**A**,**C**) or ME2 shRNA (**B**,**D**) were treated with dimethyl pyruvate (1 mM) for 48 h. Migratory properties were analyzed using wound healing assays. The statistical analysis of the relative open wound of the indicated cells (left). Representative images are shown (right). (**E**–**H**), SK-Hep1 (**E**,**F**) and Huh7 (**G**,**H**) cells expressing ME2 siRNA (**E**,**G**) or ME2 shRNA (**F**,**H**) were treated with dimethyl pyruvate (1 mM) for 48 h. Migration was assessed by transwell assay. Numbers of migrated cells in the bar graph were quantified (left). Representative images are shown (right). Data in (**A**–**H**) (left) are from n  =  3 biological independent wells. Data are the mean ± s.d. Statistical significance was determined by a two-tailed unpaired *t*-test. * *p* < 0.05, ** *p* < 0.01.

**Figure 9 metabolites-13-00540-f009:**
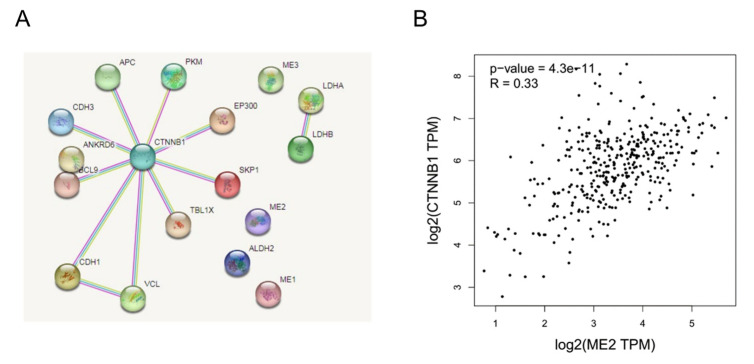
The interaction between ME2 and β-catenin. (**A**) The PPI network and clusters analysis of β-catenin. β-catenin does not interact with ME2. (**B**) Correlation analysis of ME2 and CTNNB1 (β-catenin) expression in human liver cancer patients from the GEPIA database.

## Data Availability

The data presented in this study are available in the main article.

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
