# Peer review of "ME2 Promotes Hepatocellular Carcinoma Cell Migration through Pyruvate"

_metabolites, 2023, doi:10.3390/metabo13040540_

Round 1

Reviewer 1 Report

This work entitled “ME2 promotes hepatocellular carcinoma cell migration through pyruvate” by Yanget al., they identified the role of ME2 in regulating liver cancer cells migration via increasing the pyruvate production, which binds to and increases β-catenin protein stability, thereby promoting cell migration and invasion. The present study provides a mechanistic understanding of the link between ME2 and cell migration and invasion. It is professionally written, and well presented, and as such no major comments but there are some minor comments:-

  1. Figures 1, 2 and 3 is over crowed and not easy to follow, please increase the size of each data point in each figure, you may increase the figure numbers.
  2. The conclusion is very small, please include the future direction in the conclusion.
  3. In the material and method section please explain in detail such as how many cells seeds are in each plate for how long.

Reviewer 2 Report

In this article, Yang et al. showed that malic enzyme 2 (ME2) is the promoter of cancer metastasis using hepatocellular carcinoma cell lines SK-Hep1 and Huh7. They further demonstrated that ME2 regulates pyruvate levels which in turn regulates Beta-catenin protein expression. Although the analysis was carefully performed some clarifications are needed to strengthen the rationale of the study and the conclusions.

Comments are as follows:

Comments:

1. In the abstract the authors should introduce SK-Hep1 and Huh7, i.e., what kind of cells are these. Also, the very first lines (lines 11-12) of the abstract can be divided into two sentences to maintain the flow of the reading.

2. What is the function of FBP1 and LDH? Since the authors mention these enzymes, they should also clarify this jargon for the general readers.

3. For the wound healing assays the authors used the 24 hrs. timepoint. From the representative pictures, it is clear that 24 hrs. are not sufficient for complete healing for the control samples. Why did the authors choose 24 hours instead of going for 48 hrs. to see complete wound healing and the difference with the knock-down may be bigger?

4. For Figures 1G-L; there is a discrepancy in the controls. For both SK-Hep1 and Huh7, compared to knockdown controls, the number of cells became almost half with pLV (overexpression) controls. Although the systems are different, there should not be a huge difference in the controls themselves. What is the explanation for this discrepancy?

5. For Figures 2A and 2B the authors should show the levels of ME2 knockdown by qRT-PCR.

6. In lines 284-285, the authors mentioned their previous study with acetaldehyde dehydrogenase 2 (ALDH2) without explaining why this is relevant to this study and whether ALDH2 has any function in hepatocellular carcinoma since like beta-catenin it’s also the target of pyruvate.

Reviewer 3 Report

The authors' aim is to show the role of malic enzyme 2 (ME2) in HCC cell migration and invasion. They measured cell migration and invasion in hepatocellular carcinoma (HCC) model cell systems as the main observable. They report that ME2 depletion and over-expression reduce and increase tumor cell migration, respectively. They state that ME2 promotes the production of pyruvate, which, by interacting with beta-catenin (CTNNB1) and increasing its levels, acts as a chemical mediator of the functional activities of ME2.

The data seen in the light of the existing literature are not new, nor do they appear specific to cellular models of HCC.

ME2, pyruvate and beta-catenin are actors of a complex metabolic landscape in which the aggregation, arrangement and binding of an extensive set of components form the Beta-catenin-TCF complex.

ME2, pyruvate and CTNNB1 (beta-catenin) do not act individually, nor do they have direct interactions. They are actors of a complex metabolic landscape where the formation of complex, arrangement and bonding together of a vast set of components form the Beta-catenin-TCF complex.

CTNNB1 interacts physically with many other proteins (BCL9, VCL, CDH3, TBL1X, CRBBP, APC, AXN1, CDH1 SKP1, EP300, LDHB, PKM) forming complexes (see Interact and/or BioGrid, also for the references) activating innumerable properties functional. CTNNB1 indirectly interacts with ME2, and the relationships between these two proteins are mediated by LDHA, LDHB, PKM and ALDH2, and through the LDHB/CTNNB1 and PKM/CTNNB1 physical complexes (Rosenbluh J et al. 2016) (see also BioGrid and /or Intact).

The logical conclusion is that ME2 and CTNNB1 do not operate directly but through a unique molecular network formed by protein complexes where they collectively implement functional biological processes such as Wnt signaling pathway (p value = 9.8e-9), Cadherin binding (2.9e-4), Beta-catenin destruction complex (1.4e-5), Lactate catabolism (1.1e-5), Glycolisis (3.0e-4), Pyruvate metabolism (2.9e-9), Pathways in cancer (5.8e-6), Glucagon signaling path (1.5e-6), Central carbon metabolism in cancer (6.7e-5), miRNA and lncRNA regulation in Wnt signaling. These functional actions occur in many tissues, including liver (2.0e-5)) and cancerous tissues.

Systems Biology explains that no information relating to only 2 actors of the system can be extracted from such a complex network of interacting actors without knowing the behavior of the other actors, simply because they operate together.

Certainly ME2 and pyruvate and CTNNB1 are involved in cancer cell migration and invasion, but they cannot be interpreted according to the authors' hypotheses until we know what the complete system is actually doing. This is not the correct approach. In these cases, an interactomic approach, or Gene Network Analysis, or Gene Regulatory Networking, is more appropriate, because complementary to the found results, but capable of quantitatively explaining the role of the individual actors.

In the current context, the effects that are seen are macroscopic and can originate from different profound mechanisms, but we have no information on what is happening at the overall molecular level.

That ME2, pyruvate and CTNNB1 behave functionally similarly across the different cancer types studied suggests that their role is common to all cancers and is integrated into the network (Yung-Lung-Chang et al., 2014; Pongratz RL, et al., 2007; Ren JG et al., 2014; Yang C. and Hensley, 2014;).

The authors omit much of this literature which they should comment on because it reports effects common to a good deal of cancers that are essentially similar. Why should HCC be different? If so, it must be proved irrefutably.

Round 2

Reviewer 2 Report

The authors addressed my concerns. The manuscript can be accepted. 

Author Response

We thank the reviewer for taking the time to review our manuscript and for the positive comments.

Reviewer 3 Report

The authors have changed nothing in their manuscript, except for a few sentences. The experiment carried out confirms what is already known.

ME is present in 3 isoforms (ME1, ME2, ME3) which regulate pyruvate metabolism. Regarding the role of individual enzymes, in the literature, they are reported as different and involved in various forms of cancer.

The authors state in several places:

“Over-expression of ME2 has also been associated with cell migration and invasion [17, 18].

ME2 is involved in TCA cycle and regulates glutamine metabolism in some cancer cells [14, 28].

ME2 plays an important role in regulating cell growth, proliferation, and invasion of a variety of tumor cells [14, 17, 18, 29].

In addition, ME2 also regulates insulin secretion in INS-1 cells through the pyruvate cycling [30].

ME2 plays a crucial role in modulating lung cancer differentiation and growth [31].

In this study, we show the role of ME2 in promoting the migration and invasion of hepatocellular carcinoma cells. Notably, this function of ME2 is mediated by its product pyruvate, which directly binds to β-catenin and increases β-catenin protein levels.

Our study reveals a previously unrecognized and metabolically independent role for pyruvate in the control of cell migration and invasion by acting as an endogenous regulator of β-catenin.”

Pyruvate is a small metabolite and its involvement with catenin needs to be studied using a metabolomics approach. However, the physical or functional relationships between the ME family and the catenin, whether direct or indirect, need to be studied through an interactomic approach.

I had suggested that the authors do a quick interactomic analysis to place their hypothesis in what is known globally about the molecular relationships of catenin in human and cancer metabolism.

The authors replied it was a heavy task to carry out in the future.

I did the analysis in one morning.

Interactomics is an analytical tool that is indispensable today in any study involving the molecular relationships (physical and functional) between molecules. It's also useful just to get an idea that what you're studying makes sense in relation to human knowledge.

Simply put, the interactome highlights the known relationships between molecules and can be brought, by enrichment, to the limits of human knowledge of the field. What you need to pay close attention to is the statistical significance of what you will be analyzing. Unfortunately, in recent years, there has been a strong increase in statistically insignificant or not at all significant data, virtual data, which create scientific virtual bubbles because they are also present, for example, in PubMed, where they are cited and re-cited several times up to appear as if they were real experimental data, creating distortions in scientific designs. It is a type of pollution that has many origins that are difficult to eliminate. An interactomic analysis with a high statistical significance highlights these problems and allows to have correct information on which to carry out suitable experiments and correct hypotheses.

 Attached is a PDF file with my analysis.

Round 3

Reviewer 3 Report

I appreciated the tenacity and expertise of the authors who allowed them quick and pertinent answers with the help of new experiments. I agree with their conclusions. Now these conclusions are even more clear to readers.

Although it is a powerful technique, the interactomics analysis always operates within the perimeter of human knowledge, similar to the artificial intelligence. When knowledge about an event is low, the analysis does not extract the relationships. But there are also functional relationships deriving from ill-proven hypotheses. This generates pollution of real scientific knowledge (that based on experiments), producing "virtual knowledge" which also flows into the networks, polluting them. That's why you have to be very strict in the parametric settings of a network.

If there is a relationship between ME2 and CTNNB1, this is indirect and, at the moment, the most significant hypothesis is that there is an active role of pyruvate.

I just want to suggest to the authors my personal consideration resulting from the analysis I had to make for the revision of their manuscript.

There is indirect evidence for ME2, LDHA and LDHB suggesting a functional link among these three proteins. It concerns the behavior of putative homologs also from other organisms (PMID:32135007: Light-responsive expression atlas reveals the effects of light quality and intensity in Kalanchoe fedtschenkoi, a plant with crassulacean acid metabolism. Zhang J, et al., Gigascience. 9 (3) 2020). Or even, (PMID:31881713: TCA Cycle Rewiring as Emerging Metabolic Signature of Hepatocellular Carcinoma. Todisco S, et al., Cancers (Basel). 12(1) 2019).

There is also indirect and direct evidence of interactions between LDHA and JUN and between LDHA and EP300 suggesting functional links: a) in the Pathway Interaction Database (https://cgap.nci.nih.gov/Pathways/BioCarta_Pathways) b) in INTACT, from (PMID:21620138: Pyruvate kinase M2 is a PHD3-stimulated coactivator for hypoxia-inducible factor 1. Luo W, et al., Cell. 145(5):732-44 2011); c) from crystallography (PMID:12437352: Structurally distinct modes of recognition of the KIX domain of CBP by Jun and CREB. Campbell KM, et al., Biochemistry. 41(47):13956-64 2002).

There is evidence again in the Pathway Interaction Database of functional interaction between JUN and CTNNB1 and in BioGrid between EP300 and CTNNB1 (PMID:30836930: Convergence of Canonical and Non-Canonical Wnt Signal: Differential Kat3 Coactivator Usage. Lai KKY , et al., Curr Mol Pharmacol. 12(3):167-183 2019).

This information indirectly links ME2 to CTNNB1. Maybe this information can be useful to you in solving your research problems.

Good luck

Author Response

       We feel great thanks for reviewer’s professional review work on our article. We are grateful to the reviewer for the positive comments and constructive suggestions on our work. According to your nice suggestions, we have added the discussion and cited several references. Please see page 14 of the revised manuscript, line 359-362.

       Thank you so much once again for your extremely helpful suggestions. They make our conclusions are clearer to readers. We would like to express our gratitude to you for helping us make this a much better paper.